# Indigenous Health Research Mentorship within Post-Secondary Institutions in Canada, the United States, Australia, and New Zealand: A Scoping Review

**DOI:** 10.3390/ijerph20216973

**Published:** 2023-10-25

**Authors:** Anita Manshadi, Krista Stelkia

**Affiliations:** 1Centre for Collaborative Action on Indigenous Health Governance, Simon Fraser University, Burnaby, BC V5A 1S6, Canada; krista_stelkia@sfu.ca; 2Faculty of Health Sciences, Simon Fraser University, Burnaby, BC V5A 1S6, Canada

**Keywords:** Indigenous mentorship, Indigenous health research, post-secondary institutions, Indigenous health and wellness

## Abstract

Indigenous peoples have been engaged in research since time immemorial, and have always acknowledged the power of their own knowledge systems, ways of being, and approaches. However, Indigenous peoples continue to be underrepresented in health research within academic institutions. There is an increased need for Indigenous leadership in health research, including greater Indigenous autonomy, mentorship, and self-determination in health research. This scoping review aims to explore Indigenous mentorship within Indigenous health research in post-secondary institutions in Canada, the US, New Zealand, and Australia. A review of empirical studies, case studies, reviews, commentaries, and grey literature was conducted. Four databases were used: Web of Science, PubMed, Native Health, and Google Scholar. Out of 1594 articles, 11 articles met the inclusion criteria. Four overarching themes were identified: (1) reciprocity: giving back to community; (2) supporting the development of research skills to build research capacity; (3) fostering a sense of belonging; and (4) building student ownership and confidence. The findings suggest that Indigenous mentorship is vital to creating supportive research environments for Indigenous students in the area of health sciences. Indigenous mentorship holds promise to address challenges faced by Indigenous scholars within post-secondary institutions, including intellectual, social, and cultural isolation, and can help to foster greater integration of Indigenous worldviews in Western-dominated academic settings and research systems. Future research should examine place-based mentorship opportunities for Indigenous students in community-based health research environments. Fostering Indigenous mentorship in health sciences is essential for advancing the health and wellbeing of Indigenous peoples and communities.

## 1. Introduction

Indigenous peoples have been leading and conducting research since time immemorial and have always acknowledged the power of their own knowledge systems, ways of being, and approaches [1,2]. As such, many Indigenous health researchers, and communities locally as well as globally recognize the urgent need to transform and reclaim distinct community research processes. However, Indigenous peoples and communities carry deep roots of mistrust of health research and refer to research as the “dirtiest” word due to historical and ongoing negative and exploitative experiences with Western-based researchers that have resulted in stolen intellectual property, harm, and unethical experiments being conducted without consent [3,4]. Further, researchers have often been criticized for culturally unsafe research practices including: (1) unethical research engagement; (2) reporting on the health of Indigenous populations from a deficit-based perspective by highlighting negative health outcomes over strengths- and rights-based approaches; (3) lack of Indigenous representation and input in validating research findings and knowledge translation; and lastly, (4) failing to return research findings to the community(ies) [1,5,6]. Moreover, Indigenous knowledges, perspectives, and community-driven research practices in health research have not prominently been honored or respected in a culturally safe manner within Western-based research environments, including academic institutions [7].

There is increased recognition of the need for Indigenous leadership in health research, and the need for greater Indigenous autonomy, mentorship, and self-determination in health research [7,8]. Furthermore, the Truth and Reconciliation Commission Calls to Action #23 highlights the critical need for Indigenous representation in health professions, specifically in healthcare fields [9]. Currently, Indigenous students, Indigenous health researchers (IHRs), and faculty members across Canada, the United States (US), Australia, and New Zealand are underrepresented in Western-dominated academic institutions, as well as in health research [10,11]. In Canada, Indigenous students constitute 5% of all post-secondary students, and Indigenous faculty members represent only 1.4% of all faculty members [12]. According to the 2012 US Census Bureau, less than 1% of Native American students in the US earn a bachelors, graduate, doctoral or professional degree [13]. In 2018, there were 18,062 Indigenous post-secondary students, making up 1.8% of the post-secondary student population in Australia [14]. There is a body of literature that articulates the underrepresentation, retention, and retainment of Indigenous students, early career researchers (ECRs), IHRs and faculty members in academia, due to structural racism, systematic challenges, and the intellectual constructs of colonial ways of teaching and curricula [7,10]. Overarchingly, there is a common thread which speaks to the dire need for Indigenous mentorship within academic settings [15].

Indigenous mentorship is a cultural custom that has existed for centuries and can be referred to as a non-hierarchical approach in which mentors and mentees honor each other’s knowledges and gifts [7,16]. The principles of mutual recognition, respect, sharing, trust, and mutual responsibility facilitate a wholistic relationship of reciprocity between mentors and mentees [10,15]. Indigenous mentorship focuses on building relationships that are respectful and reciprocal, and that recognize each other’s sociocultural contexts in ways that allow for vulnerability, trust, and confidence to emerge [17]. Through these relationships, mentees are guided to maintain cultural integrity, find their gifts and community roles, and understand their responsibility of reciprocity [18]. 

Indigenous mentorship in post-secondary institutions is identified as a critical component in not only the academic and personal growth of students, ECRs, IHRs, and faculty members, but also facilitates a richness of cultural wealth, identity, and supports confident Indigenous health leaders to actively pursue health research and projects that would benefit their community(ies) [8,15,19]. In Canada, the creation of Indigenous mentorship programs such as the Launching Native Health Leaders (LNHL), Indigenous Capacity and Development Research Environments (ICDRE), and the Network Environments for Indigenous Health Research (NEIHR) are examples of catalysts that not only provide meaningful mentorship spaces for Indigenous students, scholars and ECRS, but also increase Indigenous representations in health and research fields [7,20].

This scoping review explores the gaps, experiences, and opportunities of Indigenous mentorship among self-identified Indigenous post-secondary students and research trainees who are interested in Indigenous health research. Additionally, the relationship and kinship among the mentors and mentees within Western institutionalized post-secondary environments and research capacity building will be explored. To date, there has been limited attention on Indigenous mentorship within post-secondary institutions for self-identified Indigenous post-secondary students and research trainees who are interested in Indigenous health research. Therefore, this review aims to contribute towards a gap in existing knowledge by exploring Indigenous mentorship within Indigenous health research in post-secondary institutions.

## 2. Materials and Methods

The review was conducted to search and critically review peer-reviewed and grey literature using the Preferred Reporting Items for Systematic Reviews and Meta-Analyses extension for Scoping Reviews [PRISMA-ScR] as a reporting guideline. We adopted the five-step scoping review framework of Arksey and O’Malley [21], including (1) identifying the research question; (2) identifying relevant studies; (3) study selection; (4) charting the data/data extraction; and (5) analysis and reporting the findings.

### 2.1. Identifying the Research Question

The population, concepts, and context (PCC) framework was used to support the eligibility of the research question [22]. The population of interest was self-identified Indigenous post-secondary students and research trainees. The concept consists of experiences of mentorship opportunities between Indigenous and non-Indigenous Faculty, adjunct members, and Indigenous health researchers with Indigenous students and research trainees within the field of Indigenous health research in post-secondary institutions. Lastly, the context was limited to Canada, the US, Australia, and New Zealand. The primary research question was the following: *what is the current state of knowledge and opportunities in Indigenous mentorship among Indigenous post-secondary trainees and students pursuing Indigenous health research?* The secondary question this review seeks to answer is *How are Indigenous post-secondary students and research trainees being mentored by Indigenous and non-Indigenous faculty, adjunct members, and/or senior Indigenous health researchers within the field of Indigenous health research in post-secondary institutions?*

### 2.2. Identifying Relevant Studies

A comprehensive search was conducted by examining four databases including Web of Science, PubMed, Native Health, and Google Scholar. To keep the review relevant to Indigenous mentorship within Indigenous health research in post-secondary institutions, the authors focused their search timeline on reviews of published articles and reports between January 2000 and February 2023. Table 1 outlines the specific keywords used for the search strategy, which were collaboratively developed with the support of the University of Victoria’s Public Health Librarian and the second author. We included multiple terms in our search strategy to provide a broad overview and an in-depth synthesis of the literature that is reflective of distinct Indigenous groups.

Our initial inclusion criteria for selecting articles consisted of six components: (1) empirical studies, case studies, reviews (i.e., systematic reviews, scoping reviews, literature reviews, narrative reviews), commentaries, and grey literature; (2) articles written in the English language; (3) full-text, peer-reviewed journals; (4) articles published from January 2000 to February 2023; (5) articles focused on Indigenous mentorship within Indigenous health research in post-secondary institutions; and (6) articles that were published in the specified locations, such as Canada, the United States, Australia, and New Zealand.

Exclusion criteria consisted of articles that do not focus on self-identified Indigenous peoples, post-secondary students, trainees, Indigenous health researchers, faculty, and adjunct members not related to Indigenous mentorship within post-secondary institutions, not related to Indigenous health research, and not within Canada, the US, Australia, and New Zealand. Lastly, studies that were published prior to January 2000 and not in English were excluded. The inclusion and exclusion criteria were also applied to grey literature, which was screened with the same criteria as peer-reviewed articles.

### 2.3. Study Selection

A total of 1594 studies were identified in the search and imported into Mendeley, a reference management software, and then exported to Covidence, a web-based software platform for screening, data extraction and removal of duplication citations. Using Covidence, 668 duplicates were removed. A total of 926 studies were screened independently by two reviewers for the titles and abstracts, and 862 irrelevant studies were removed. The full text of the remaining 64 studies were reviewed and 53 studies were excluded based on inclusion and exclusion criteria. Any disagreements or uncertainty regarding the identified studies were resolved through ongoing discussions to achieve full consensus on the review. The final studies included in the scoping review totaled 11. Figure 1 illustrates the results of the screening process of the studies for the review, which is presented in the Preferred Reporting Items for Systematic Reviews and Meta-Analyses (PRISMA) statement.

### 2.4. Charting the Data (Data Extraction)

The first author independently extracted the data into a standardized form through Covidence to chart the data. The data in the table were assessed for consistency and for any discrepancies throughout the data-charting process. The following items were catalogued into a table: title/authors/year/country; Indigenous group; population category; sample size; goals/aim/objectives; study design and methods; name of post-secondary institution; defining Indigenous mentorship; and types of mentorships offered. Appendix A outlines the 11 studies that were explored for this scoping review.

### 2.5. Analysis and Reporting the Findings

Following an Indigenous conceptual framing, we used a thematic analysis approach to articulate the narratives presented in the literature [2]. We used a hybrid approach of inductive and deductive coding to present themes in a cohesive and holistic way that encapsulates the overall aim of this review [2,23]. The primary author (A.M.) undertook a preliminary reading of the included studies and developed initial overarching themes based on the research question, using the research questions as a guide. The primary author (A.M.) conducted inductive line-by-line coding to identify the themes for this review as well as deductive coding using the framework of the research questions. Themes were collated in a Microsoft Word document by A.M. In the second phase of analysis, the authors returned to reviewing the included studies and undertook thematic coding. A collective review of all themes, similarities, and differences was undertaken to create a geographically specific thematic approach. After the final themes were developed, A.M. clarified the themes by adding definitions and a high-level summary where necessary. Once coding was concluded, a final review of the overarching themes was discussed and validated between the authors (A.M., K.S.).

## 3. Results

### 3.1. Characteristics of the Included Studies

As depicted in Figure 1, 11 articles were identified to have met the inclusion criteria. The details of each study included in this review can be found in Appendix A. The review consisted of two empirical studies, five case studies, one commentary, and one of each of the following reviews: literature, systematic, and narrative review. The majority of the articles were published between 2008 to 2022. Their sample sizes ranged from 7 to 1067 and involved Indigenous undergraduate and graduate students, faculty members, the health researcher workforce, Indigenous mentors, early career scholars, Indigenous researchers, lead investigators, project staff members and one article with high school through to graduate and medical student populations.

From the 11 articles, distinct Indigenous groups were identified based on the geographical location. For example, there were two articles published in Canada that identified their sample population as Indigenous, the six US articles referred to Native American and Native Hawaiian, and three articles from Australia identified Aboriginal and Torres Strait Islanders as their population in the articles. Although our search strategy included New Zealand, there were no articles from this geographical location that met the inclusion criteria.

### 3.2. How Indigenous Health Mentorship Programs Support Capacity Building

The results reveal there are many pathways through which Indigenous mentorship among post-secondary students and research trainees can facilitate promising opportunities for capacity building in Indigenous health research. More specifically, there were four overarching themes that emerged from this review: (1) reciprocity: giving back to community; (2) supporting the development of research skills to build research capacity; (3) fostering a sense of belonging; and (4) building student ownership and confidence.

#### 3.2.1. Reciprocity: Giving Back to Community

The first theme identified how Indigenous health mentorship programs help to advance reciprocity through the opportunity to give back to community. Seven articles discussed how the “researched are becoming the researchers”, and the meaningful commitment to giving back to their community(ies) on health matters that are locally relevant and hold promise for positive change [7,8,10,13,24,25]. For example, one study highlighted a Native Hawaiian mentee’s perspective on her success in pursuing higher education and how in return she is positively impacting her community. She perceived this as forging a path for future generations of Native Hawaiians to pursue higher education and give back to their community with a positive narrative [10]. Another mentee shared that “coming together as students from different parts of the reservations and discussing and knowing that, okay, I go to school for a purpose. Additionally, you leave here with encouragement of making a difference when you back to your community” [6] (p. 2218). The article of Hughes et al. [13] illustrated how four undergraduate students from Diné college specified that they want to continue living and working in the Navajo Nation due to the enriching learning experiences and mentorship they received from the 10-week cancer research fellowship program facilitated by the Mayo Clinic and Diné College. One student shared “I hope to work for the Navajo Nation and help to influence positive change in my community on the reservation regarding public health” [13] (p. 97). 

Mentors expressed adopting a community-based lens in student research has not only assisted the students’ cultural identity, trust, and reciprocal relationships with communities; it has also validated, empowered, and provided culturally relevant support to students, while in return benefitting the community’s transmission of Indigenous knowledge, values, and resources [10]. A mixed-methods study discussed four Indigenous students who expressed that “we need more Natives treating Natives... I got to help my people… and that is how I ended up at the Native American Research Internship” [10] (p. 574). One student conveyed her interest in being a pediatrician and conducting research on diabetes and childhood obesity, as this is a rising health concern and cause of death in her community [25].

The findings from the seven articles share a diverse range of narratives and reflections of mentees and mentors expressing how through university-based mentorship programs, research activities, and research affiliations with Indigenous communities were facilitators in shaping mentees’ educational journey and research interests that are in alignment with their community’s needs [6,7,8,10,13,23,24].

#### 3.2.2. Supporting the Development of Research Skills to Build Research Capacity

The second theme identified in Indigenous health research mentorship programs was supporting the development of research skills to build research capacity. The majority of articles discussed building research capacity through formal and informal mentorship programs and activities for the personal and professional growth of Indigenous students, ECRs, Indigenous health researchers, as well as faculty members [6,7,8,10,13,24,26,27,28,29]. Segrest et al. [8] expressed the purpose of the LNHL program was due to the “anemic representation” of Indigenous researchers to confront Indigenous health inequities and be at tables where research agendas and policies are being discussed and created. This article aimed to explore Indigenous undergraduates participating in the LNHL program through attendance of health and research conferences facilitated by Tribal community representatives on tribal health issues. Undergraduate students convened in a safe space where the facilitators encouraged students to rediscover and weave “Native ways of knowing and learnings” when faced with Western-built academic and research environments by keeping their own identities, lived experiences, and traditional knowledge “in check” [8].

In a regionally based Australian University, an Indigenous research capacity-building intervention for Aboriginal and/or Torres Strait Islander and non-Indigenous researchers used research capacity-building tools such as writing retreats (publications, conference presentations, reports, book chapters and other grey literature) and five-day residential workshops on research skills training and professional development as another form of mentorship. The outcomes of this project and mentorship activities resulted in many Indigenous researchers formalizing new and existing mentoring relationships in their professional, academic, research and community networks [24]. Additionally, there was an increase in publications from the Indigenous researcher participants who were involved in the writing retreats and research skills training workshops [24].

Within Canada, the use of mentorship networks such as the ICDRE and NEIHR programs led to a transformative impact in the development of a national cadre of scholars in Indigenous health scholarship [7]. Specifically, the Ontario Indigenous Mentorship Network has established an ecosystem of Indigenous health research for 13 Ontario universities, among which more than 70 research trainees, ECRs, Indigenous researchers and community advisors have participated in various mentoring sources, activities to empower the training needs and career trajectories of Indigenous health students, and ECRs [7].

In the US, a 10-week Native Cancer Research Experience and Student Training (CREST) program for Navajo undergraduate students at the Mayo Clinic Cancer Centre and Diné College introduced several mentorship opportunities at the grassroots level [13]. Students were active team members on cancer-related projects and were encouraged to take a bicultural approach to research activities while integrating Indigenous knowledge and Western biomedical knowledge in their research practice. Seven students from the CREST program felt empowered with the practical knowledge and research skills offered by their mentors and increased interests in community-based research projects on the Navajo Nation and in other Indigenous communities [13]. Furthermore, Indigenous undergraduate students collaborating with a faculty mentor on hands-on research projects increased the students’ research skills, self-efficacy, confidence, and enhanced their interest in further pursuing graduate studies [10].

Additionally, another study addressed findings from a survey that 59% of respondents (present and past researchers in Indigenous health) felt their education, specifically their lack of exposure to Indigenous health issues, inadequate research and cultural training, and the absence of mentorship, did not prepare them to work in the field of Indigenous health research. Despite the respondents’ challenges due to their lack of exposure to Indigenous health and a lack of research training and mentorship, 45% of respondents would recommend a career in Indigenous health research to an ECR [28].

#### 3.2.3. Fostering a Sense of Belonging

Fostering a sense of belonging was identified as a theme among the studies as a way to contribute towards enhancing the mentees’ learning experiences, academic performance, sense of purpose, scientific and cultural identity, and kinship with peers [7,8,10,24,26]. One case study shared the perspective of five different Indigenous communities (Chippewa-Cree, Native Hawaiian, the Nome Eskimo Community, Tsimshian, and the Absentee Shawnee Tribe of Oklahoma) and described how the Native Children’s Research Exchange (NCRE) Scholars program helped address the intellectual and social isolation in academic and community settings through a collaborative interdisciplinary network of senior Indigenous health researchers, ECRs, and graduate students [26]. This relational bridge fostered a sense of belonging for Indigenous students and emerging ECRs, whether they were engaging in Indigenous health research locally or globally. A PhD graduate shared “I have richly benefitted from mentorship with other NCRE network researchers who have shared insights and perspectives with me from their own experience as early career Native scientists” [26] (p. 7). At the heart of Ontario’s Indigenous Mentorship program is the Anishinaabe philosophy *Mno Mimkodadding Geegi:* “we are all connected” [6]. One commentary described that the program not only provided financial means for trainees and students while pursuing their education, it also provided a sense of belonging for new cohorts of Indigenous health trainees. 

Chow-Garcia et al. [25] found that mentors played a significant role in facilitating Indigenous students’ sense of belonging in science. Students affirmed the power of relationship building and belonging with their mentors and fellow peers. For example, a student shared “I was supported by other grad students that I met there, and the mentors and directors of the program really wanted me to succeed and they—we still keep in contact, so they still really care…” [25] (p. 576).

Indigenous mentors practicing relationalism by embedding traditional protocols and practices, such as sharing circles, gifting, smudging, or reflexive introductions in academic and community settings, created safer spaces and a sense of belonging for mentees in Western built academic and research environments [24,25,27]. Overall, five articles highlighted that culturally tailored mentorship models, activities, and programs provided a sense of belonging to the students, giving them a research community without disrupting their cultural identities and Indigenous worldviews [8,10,25,26,27].

#### 3.2.4. Building Student Ownership and Confidence

Student ownership and confidence was revealed to be a pivotal and common theme throughout the 11 articles reviewed. Zorec [10] explored the concept of self-directed learning, in which students become independent critical thinkers and take ownership of their research. One mentee described his research process with his mentor: “It’s not like she is going to be walking me through every single step along the way. A lot of it is giving me the tools to go on by myself, being independent working by myself… (p. 8)”. The findings revealed that mentees showed greater leadership, self-initiative, and confidence in their research approach and projects [10]. Cultivating reciprocal relationships with Indigenous communities by communicating Indigenous health research to communities was an example of mentees expressing their confidence, self-efficacy and ownership in building culturally safe research relationships [10]. One mentee described her self-confidence as follows: “[I] make my own relationships [with communities]. I go out and meet new people without her [the mentor], and it was because of what she taught me (p. 9).”

There were several positive qualitative outcomes that transcended from the 11 articles that explored building student ownership and confidence. Two mentees from the LNHL and CREST program expressed that “this session offered a framework for understanding the purpose of my own journey. I became confident in my chosen pathway [8] (p. 83).” The second mentee from the 10-week cancer research training program confidently shared that “I was able to claim ownership of my project and I was required to defend it myself within an academic setting (p. 96).”

## 4. Discussion

The findings from this review suggest there are many positive benefits to Indigenous mentorship through diverse pathways including, academic mentorship programs and activities (i.e., writing retreats, virtual writing workshops, health research conferences, intensive research training programs) as well as community-based research projects with a partnered Indigenous community. Seven out of the 11 articles provided recommendations for enhancing Indigenous mentorship within academic and community settings by promoting culturally grounded mentorship activities and programs that can help build student success, scientific and cultural identity, a sense of belonging, and confidence in students’ ability to conduct research within their own communities [6,7,10,13,25,26,28]. Most of these studies also acknowledge the challenges that Indigenous students, ECRs, and Indigenous faculty members face in Western-based academic institutions and health research systems.

The findings from this review align with the existing literature, which has found that adopting traditional protocols and practices such as storytelling, talking circles, and smudging [9,27] empowered Indigenous mentees to use their traditional knowledge and values as strengths. This enhanced their self-efficacy and scientific and cultural identity in Western-based academic institutions and health research systems [7,8]. There is a critical need for Western-based academic institutions and health research systems to adopt and honor Indigenous knowledge systems, respectfully engage and build authentic research relationships with Indigenous communities, and increase the representation of Indigenous health scholars and students [21]. This transformative action will create opportunities wherein Indigenous students, ECRs, IHRs, and faculty members can mobilize and share their ways of knowing, cultural values, and lived experiences among their peers. This may result in a sense of belonging, developing research skills, and validating their sense of purpose, research identity, and skills. Building research capacity with and for future Indigenous health scholars through distinct mentorship can promote culturally safer health and research environments, and reciprocate positive health outcomes for Indigenous peoples and communities [7,10,13,26,27].

A promising finding of this review was the growing interest of Indigenous (i.e., undergraduate, graduate, and PhD) students and ECRs wanting to actively participate in community-based research that not only benefits their research interests but also meaningfully contributes to their community’s health and wellness [8,10,13,27]. This positive response from future Indigenous health scholars and leaders is a driver for conducting impactful research with interdisciplinary teams who can provide support in developing research skills and methodologies, but can also determine their own futures, thereby contributing to self-determination in health research and Indigenous sovereignty [7]. Additionally, the findings of this review highlight the extent and range of the available literature on this important topic in Indigenous mentorship in health research and suggest areas that require further research.

Overall, the findings reveal that Indigenous mentorship has positive benefits for helping to improve and advance the health and wellbeing of Indigenous peoples and communities. Indigenous mentorship provides a pathway for promoting mental, emotional, spiritual, and physical health and wellbeing, as well as resiliency, healthy relationships with kin and cultural continuity by fostering inclusion. Furthermore, by mentoring the next generation of Indigenous researchers and scholars, the investment of time, skill development and relationship development will help to facilitate a stronger network for Indigenous peoples to lead research that is community-engaged, self-determining, and aligned with Indigenous worldviews. Therefore, supporting Indigenous students to become future researchers and scholars will help to better support Indigenous-led research that will have a meaningful impact on overall Indigenous health and well-being.

### 4.1. Future Research

Future research on Indigenous mentorship among Indigenous post-secondary students and ECRs who are interested in pursuing Indigenous health research is required. Firstly, future reviews should examine peer-reviewed and grey literature in other regions of the world in which Indigenous people live, such as Central and South America, Asia, and Africa. Exploring mentorship models in these regions will help to further expand our knowledge on the topic. Secondly, it is recommended that future studies examine place-based mentorship activities for post-secondary students and ECRs in community-based health research. This will result in more context-specific findings that will contribute towards a greater understanding of mentees’ research interests, building relationships with Indigenous communities and addressing Nation-specific health inequities and outcomes. Additionally, future studies should consider exploring Indigenous mentorship in other fields such as community health, nursing, science, and Indigenous knowledge on the environment and sustainability. This may help to better enhance Indigenous mentorship and build research capacity within specific fields.

### 4.2. Limitations

This review has several limitations that should be considered. The first limitation is the review was limited to peer-reviewed and grey literature that was published in English. Therefore, Indigenous mentorship articles written in different languages were not included as part of our analysis. Secondly, the geographical scope of this review was limited to Canada, the United States, Australia, and New Zealand; thus, it does not include articles on Indigenous mentorship that were published in other regions of the world in which Indigenous peoples live, such as Central and South America, Asia, and Africa. Additionally, we only reviewed empirical studies, case studies, reviews (systematic, scoping, narrative, literature, commentaries), and community reports that were published between January 2000 and February 2023. Furthermore, since our review focused solely on Indigenous mentorship in the field of Indigenous health research within post-secondary institutions, our study findings do not reflect the perspectives and experiences of Indigenous students, trainees, and ECRs mentorship experiences in community settings involved in Indigenous and community-driven research projects. Another limitation of this review was restricting our search to Indigenous health research and not broadening the topic to Indigenous health, community health, nursing, and science.

## 5. Conclusions

This scoping review aimed to understand and map existing body of literature in Indigenous mentorship in relation to Indigenous health research within post-secondary institutions, and to identify areas for future research. Our review included 11 articles, and four overarching themes were identified: (1) reciprocity: giving back to community; (2) supporting the development of research skills to build research capacity; (3) fostering a sense of belonging; and (4) building student ownership and confidence. While Indigenous mentorship remains a promising component to addressing the complexity of contemporary Indigenous health inequities, structural racism and systemic challenges, and the underrepresentation of Indigenous health researchers and leadership in health research, there is a greater need for place-based Indigenous mentorship designed specifically to align with the research interests of future Indigenous post-secondary students and emerging ECRs. Future research into Indigenous mentorship may be a catalyst for researchers, community representatives, and academic institutions to take actionable steps in supporting Indigenous mentorship, while building research capacity for a brighter future with and for Indigenous peoples and communities.

## Figures and Tables

**Figure 1 ijerph-20-06973-f001:**
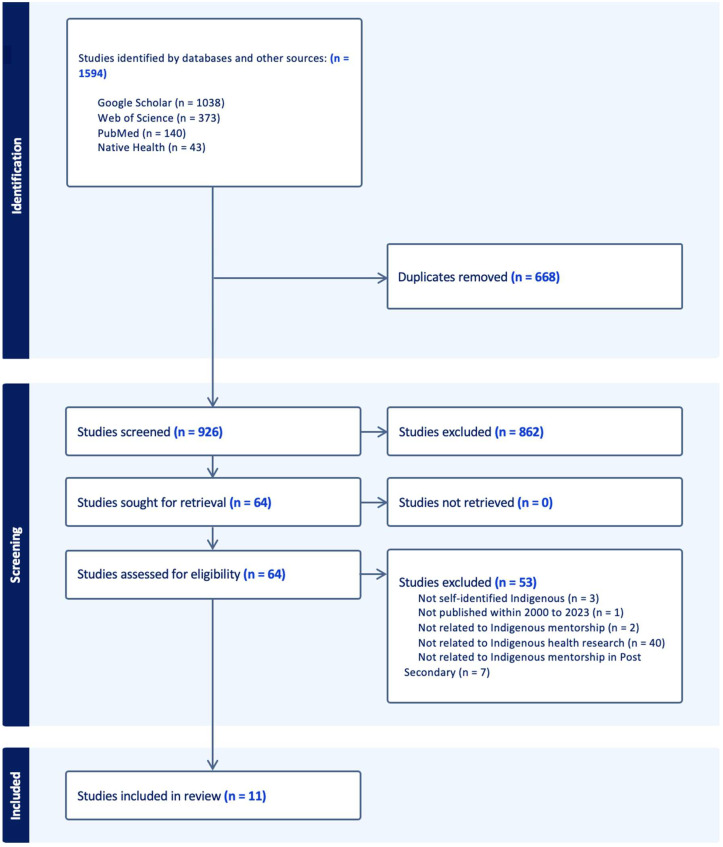
Flow diagram of included studies.

**Table 1 ijerph-20-06973-t001:** Keyword search strategy.

Concept	Keywords	Databases and Search Engines
Indigenous	“Indigenous” OR “Aboriginal” OR “First Nations” OR “Métis” or “Inuit” OR “Torres Strait Island” OR “Māori” OR “Native American” OR “Native Hawaiian”	- Web of Science - PubMed - Native Health - Google Scholar
Mentorship	“Mentorship” OR “Mentor” OR “Mentoring” OR “Mentee”	
Researcher	“Researcher*” OR “Research” OR “Scholar*”	
Post-Secondary	“Universit*” OR “College*” OR “Postsecondar*” OR “Post Secondar*”	
Country	“Canada” OR “Australia” OR “New Zealand” OR “USA”	

## Data Availability

No new data were created or analyzed in this study. Data sharing is not applicable to this article.

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
