# Peer review of "Indigenous Health Research Mentorship within Post-Secondary Institutions in Canada, the United States, Australia, and New Zealand: A Scoping Review"

_ijerph, 2023, doi:10.3390/ijerph20216973_

Round 1

Reviewer 1 Report

Comments and Suggestions for Authors

I am impressed with both the quality of the writing, and also the thoughtful introduction and background which did a very nice job of laying the foundation for understanding the colonial impacts that have permiated Indigenous peoples for generations. Indigenous mentorship with in academia is an essential part of supporting success among Indigenous researchers; however to date has received little attention. I applaud the authors for addressing this gap. My only suggestion is to strengthen the discussion and weave in recommendations (minor revision) and potential discussion related to next steps and best practises for Indigenous engagement of students and faculty within the university setting. 

Author Response

Reviewer 1 comments: 

Comment 1: I am impressed with both the quality of the writing, and also the thoughtful introduction and background which did a very nice job of laying the foundation for understanding the colonial impacts that have permeated Indigenous peoples for generations. Indigenous mentorship within academia is an essential part of supporting success among Indigenous researchers; however, to date has received little attention. I applaud the authors for addressing this gap. My only suggestion is to strengthen the discussion and weave in recommendations (minor revision) and potential discussion related to next steps and best practises for Indigenous engagement of students and faculty within the university setting.  

Response: Thank you for your thoughtful response and suggestions for weaving in recommendations or potential next steps for Indigenous engagement of students and faculty within post-secondary settings. However, this review solely focused on Indigenous mentorship within Indigenous health research, I do foresee myself in the near future submitting a manuscript that discusses promising practices for Indigenous engagement of stakeholders (students, trainees, faculty, researchers, and communities) within post-secondary settings.

Reviewer 2 Report

Comments and Suggestions for Authors

This is an interesting paper on an important subject, particularly if we are to engagement with Indigenous communities on public and environmental health.

I note that of nearly 1600 articles found, only 11 met the inclusion criteria.

Initially, I found some confusion between the concepts of Indigenous Health Research and Indigenous Health Researchers but am assuming that IHR only refers to the latter.  Similarly, I was not clear initially whether the inclusion criteria meant that reviews were only included if mentors (as well as mentees) were members of Indigenous groups and if only research on indigenous health was included?  For example, if the search found review papers that included mentorship of health research on general health topics (but undertaken by Indigenous health researchers) would these have been included.

However, the fifth inclusion criterium suggests mentors were only included if they were indigenous and research topics had to be on indigenous health.

While I can see the argument for this criterium but it could mean that a lot of examples of mentorship of IHRs may have been excluded as being in the 861 irrelevant studies and 40 of the 53 studies assessed.  This is highlighted in the limitations section.

I think the discussion on limitations of this review is reasonable but could also mention the limited numbers of researchers studied (numbers of mentees and mentors are not clear) and a search over a 20-year period, when both understanding and methodologies for engagement will have evolved, might makes it more difficult to extrapolate findings to solutions.  The restriction to indigenous health research has obvious limitations and in addition to considering future reviews including community health, nursing and science, I would suggest highlighting Indigenous knowledge on environment and sustainability. 

I was disappointed that few examples given of indigenous knowledge systems that were utilised in these studies.

Comments on the Quality of English Language

There are some errors in the text.  For example, line 157 refers to the remaining 50 studies which clearly is incorrect.  I am assuming from the flow diagram in Figure 1 that it should read 64 studies.  Also line 177 has wholistic which in UK English is spelled holistic.

Author Response

Reviewer 2: 

This is an interesting paper on an important subject, particularly if we are to engagement with Indigenous communities on public and environmental health. I note that of nearly 1600 articles found, only 11 met the inclusion criteria. 

Comment 1: Initially, I found some confusion between the concepts of Indigenous Health Research and Indigenous Health Researchers but am assuming that IHR only refers to the latter.  Similarly, I was not clear initially whether the inclusion criteria meant that reviews were only included if mentors (as well as mentees) were members of Indigenous groups and if only research on indigenous health was included?  For example, if the search found review papers that included mentorship of health research on general health topics (but undertaken by Indigenous health researchers) would these have been included. However, the fifth inclusion criterium suggests mentors were only included if they were indigenous and research topics had to be on indigenous health. While I can see the argument for this criterium but it could mean that a lot of examples of mentorship of IHRs may have been excluded as being in the 861 irrelevant studies and 40 of the 53 studies assessed.  This is highlighted in the limitations section. 

Response: Thank you for your response, under “Section 2.1 Identifying the research question” I have outlined the Population, Concept and Context (PCC) framework which provides a breakdown of what was included in this review. For example, Lines 114 to 119, states “The population of interest were studies of self-identified Indigenous post-secondary students and research trainees. The concept consists of narratives and experiences of Indigenous mentorship opportunities between Indigenous/non-Indigenous Faculty, adjunct members, and Indigenous health researchers with Indigenous students and research trainees within the field of Indigenous health research in post-secondary institutions."

Comment 2: I think the discussion on limitations of this review is reasonable but could also mention the limited numbers of researchers studied (numbers of mentees and mentors are not clear) and a search over a 20-year period, when both understanding and methodologies for engagement will have evolved, might make it more difficult to extrapolate findings to solutions.  The restriction to indigenous health research has obvious limitations and in addition to considering future reviews including community health, nursing and science, I would suggest highlighting Indigenous knowledge on environment and sustainability.  

Response: Thank you for suggesting including areas of topic such as community health, nursing and science, and Indigenous knowledge on environment and sustainability. I have included the following sentence in Lines 434 to 437, “Additionally, future studies should consider expanding areas of interest in relation to Indigenous mentorship such as community health, nursing and science, Indigenous knowledge on the environment and sustainability.” 

Comment 3: I was disappointed that few examples given of indigenous knowledge systems that were utilised in these studies. 

Response: We too acknowledge the limited references of Indigenous knowledge systems in the studies that we reviewed in our scoping review. However, this is an opportunity to identify  

Comment 4: Comments on the Quality of English Language: There are some errors in the text.  For example, line 157 refers to the remaining 50 studies which clearly is incorrect.  I am assuming from the flow diagram in Figure 1 that it should read 64 studies.   

Response: Thank you for catching the discrepancies in the total number of remaining studies. I have revised line 157 to the correct number which is 64 remaining studies that were reviewed not 50. 

Comment 5:
Also line 177 has wholistic which in UK English is spelled holistic. 

Response: Thank you, I have changed Line 177 from “wholistic” to “holistic”  

Reviewer 3 Report

Comments and Suggestions for Authors

The study is timely and well-written. I only have very minor comments.

Line 25 in the abstract, you do not need to comma after including. It will read better "including identity and Indig..."

Lines 55-58 in the introduction. The sentence that starts with "Furthermore, the Truth and Reconciliation ..." I suggest you ass citations of the USA, New Zealand and Australia that also highlight the critical need for Indigenous representation in the health profession, etc. You are making your case for these nations.

Lines 67 to 72. The last two sentences of this paragraph: "While there is .... Concluding, a common thread..." They do not read well, and they are very important in your argument.

Section 2.3. Study Selection. Line 156 indicates 861 irrelevant studies, while Figure 1 indicates 862. Similarly, line 157 indicates 50 remaining studies, but Figure 1 indicates a different number.

Well done!

Author Response

Reviewer 3 comments: 

The study is timely and well-written. I only have very minor comments. 

Comment 1: Line 25 in the abstract, you do not need to comma after including. It will read better "including identity and Indig..." 

Response: Thank you for critically reviewing our paper, I have removed the comma after the word including in the Abstract on Line 25. 

Comment 2: Lines 55-58 in the introduction. The sentence that starts with “Furthermore, the Truth and Reconciliation…” I suggest you ass citations of the USA, New Zealand and Australia that also highlight the critical need for Indigenous representation in the health profession, etc. You are making your case for these nations.  

Response: Thank you for your comment. The sentence “Furthermore, the Truth and Reconciliation Commission Calls to Action #23 highlights the critical need for Indigenous representation in health professions, specifically in healthcare fields” is in specific reference to Canada solely as it speaks specifically to the Truth and Reconciliation Commission of Canada’s Calls to Action. Therefore, adding USA, New Zealand, and Australia citations to this sentence would not be appropriate given the Canada-specific context of this sentence. 

Comment 3: Lines 67 to 72. The last two sentences of this paragraph: “While there is …. Concluding, a common thread…” They do not read well, and they are very important in your argument.

Response: Thank you for your suggestion, I have revised lines 67 to 73 to say, “ There is a body of literature that articulates the underrepresentation, retention, and retainment of Indigenous students, early career researchers (ECRs), IHRs and faculty members in academia, due to structural racism, systematic challenges, and the intellectual constructs of colonial ways of teaching and curricula (Richmond, 2018; Zorec, 2022). Overarchingly, there is a common thread which speaks to the dire need for Indigenous mentorship within academic settings (Trudgett et al., 2021).”

Comment 4: Section 2.3. Study Selection. Line 156 indicates 861 irrelevant studies, while Figure 1 indicates 862. Similarly, line 157 indicates 50 remaining studies, but Figure 1 indicates a different number. 

Well done! 

Response: Thank you for identifying the discrepancies on Lines 156 and 157. I have changed line 156 to “862 irrelevant studies were removed”, as that is what Figure 1 states. I have also changed line 157 to the correct number of “64 remaining studies that were reviewed”, not 50.